# The Role of Hemodynamics through the Circle of Willis in the Development of Intracranial Aneurysm: A Systematic Review of Numerical Models

**DOI:** 10.3390/jpm12061008

**Published:** 2022-06-20

**Authors:** Yuanyuan Shen, Rob Molenberg, Reinoud P. H. Bokkers, Yanji Wei, Maarten Uyttenboogaart, J. Marc C. van Dijk

**Affiliations:** 1Department of Neurosurgery, University Medical Center Groningen, University of Groningen, 9713 GZ Groningen, The Netherlands; y.shen@umcg.nl (Y.S.); r.molenberg@umcg.nl (R.M.); 2Department of Radiology, Medical Imaging Center, University Medical Center Groningen, University of Groningen, 9713 GZ Groningen, The Netherlands; r.p.h.bokkers@umcg.nl (R.P.H.B.); m.uyttenboogaart@umcg.nl (M.U.); 3Engineering and Technology Institute Groningen, Faculty of Science & Engineering, University of Groningen, 9747 AG Groningen, The Netherlands; yanji.wei@hotmail.com; 4Department of Neurology, University Medical Center Groningen, University of Groningen, 9700 RB Groningen, The Netherlands

**Keywords:** hemodynamics, circle of Willis, numerical model, intracranial aneurysms, personalization

## Abstract

*Background:* The role of regional hemodynamics in the intracranial aneurysmal formation, growth, and rupture has been widely discussed based on numerical models over the past decades. Variation of the circle of Willis (CoW), which results in hemodynamic changes, is associated with the aneurysmal formation and rupture. However, such correlation has not been further clarified yet. The aim of this systematic review is to investigate whether simulated hemodynamic indices of the CoW are relevant to the formation, growth, or rupture of intracranial aneurysm. *Methods:* We conducted a review of MEDLINE, Web of Science, and EMBASE for studies on the correlation between hemodynamics indices of the CoW derived from numerical models and intracranial aneurysm up to December 2020 in compliance with PRISMA guidelines. *Results:* Three case reports out of 1046 publications met our inclusion and exclusion criteria, reporting 13 aneurysms in six patients. Eleven aneurysms were unruptured, and the state of the other two aneurysms was unknown. Wall shear stress, oscillatory shear index, von-Mises tension, flow velocity, and flow rate were reported as hemodynamic indices. Due to limited cases and significant heterogeneity between study settings, meta-analysis could not be performed. *Conclusion:* Numerical models can provide comprehensive information on the cerebral blood flow as well as local flow characteristics in the intracranial aneurysm. Based on only three case reports, no firm conclusion can be drawn regarding the correlation between hemodynamic parameters in the CoW derived from numerical models and aneurysmal formation or rupture. Due to the inherent nature of numerical models, more sensitive analysis and rigorous validations are required to determine its measurement error and thus extend their application into clinical practice for personalized management. *Prospero registration number:* CRD42021125169.

## 1. Introduction

Intracranial aneurysms (IAs) are prevalent in 2–3% of the general population. Rupture of such an IA leads to subarachnoid hemorrhage (SAH), which has a high mortality rate of up to 40% [1]. Fifty to eighty percent of IAs remain, however, asymptomatic and do not rupture [2]. The etiology of IA formation, growth and rupture remains unclear, although hemodynamic factors are thought to play an important role [3].

Due to the current technological difficulties in assessing cerebral hemodynamics with clinical measurement, the majority of the existing investigations have been performed in terms of a numerical model. The numerical model is a widely applied technique to tackle complex hemodynamics by computational simulation of vasculature scenarios. These models are generally classified into zero-dimensional, one- dimensional, and three-dimensional models. Over the past decades, three-dimensional models (Computational Fluid Dynamics) have been extensively used to investigate local hemodynamics. These studies have mainly focused on intra-aneurysmal blood flow directly surrounding the IA and its parent artery. Hemodynamic parameters, such as wall shear stress (WSS), oscillatory shear index (OSI), and kinetic energy, have been identified as risk factors of IA formation and rupture [4,5,6,7,8,9,10,11,12,13,14]. There is however a large variability in the underlying assumptions of these models, e.g., viscosity assumption (i.e., consider blood as Newtonian versus non-Newtonian fluid), flow stability assumption (i.e., treat blood flow as laminar, transition or turbulent flow) [15,16]. Due to the potential bias introduced in these numerical models, the meta-analysis based on the computational models might be unreliable [17]. Furthermore, whilst most studies refer to the patient-specific morphology of IA and its parent artery, the interpretation of the term ‘patient-specific’ varies [16]. 

Predominately, the classic configuration of the circle of Willis (CoW) shown in Figure 1 is taken as default. This consists of a ring-shape symmetrical arterial structure with three afferent arteries (bilateral internal carotid artery and basilar artery), six efferent arteries (bilateral middle, anterior, and posterior cerebral arteries) and three communicating arteries (bilateral posterior communicating arteries and anterior communicating artery). In such a complex hemodynamic system, the flow velocity and the intraluminal pressure in the proximal and distal part of the parent artery (a.k.a. the boundary conditions of the parent artery) are also thought to be influenced by the systemic blood pressure and the configuration of the CoW [18,19,20,21]. It is impossible to discuss the local hemodynamics in IAs while leaving aside the CoW configuration. Based on a simple mathematical model, the flow rate in the CoW can be influenced by either the efferent resistance distribution or the radius of all the segments in CoW [22], implying that a qualified ‘patient-specific’ model should be taken into account the complete CoW network with proper ‘patient-specific’ boundary conditions.

While the conventional studies with numerical model majorly focused on the local hemodynamic in the IAs, the aim of this systematic review was to investigate whether hemodynamic indices of the CoW obtained from a numerical model are relevant to the formation, growth, or rupture of IAs. 

## 2. Methods

This systematic review was conducted in compliance with the Preferred Reporting Items for Systematic Reviews and Meta-Analysis (PRISMA) guidelines [23]. The review was registered in PROSPERO (CRD42021125169). 

### 2.1. Selection Criteria

In this review, ‘hemodynamics’ was defined as any physical description of the blood flow, as simulated by numerical models. ‘CoW hemodynamic simulations’ referred to the numerical model based on patient-specific CoW, or studies comparing different scenarios of CoW. Studies with either ruptured or unruptured IAs were included.

In order to include all relevant studies, case-control, case reports, and cohort studies were assessed. We considered both cross-sectional and longitudinal studies, and did not impose any restrictions regarding patients’ age or sex. Exclusion criteria were: 1. Articles written in another language than English; 2. Reviews, editorials letters, conference abstracts; 3. Animal studies; 4. Numerical models using non-personalized boundary conditions (inflow or/and outflow conditions of parent artery).

### 2.2. Search Strategy

The electronic databases MEDLINE, Web of Science, and EMBASE were searched from their inception until December 2020 to identify peer-reviewed articles describing the correlation between CoW hemodynamics and IAs based on numerical models. Search terms for circle of Willis, aneurysm, and hemodynamics were combined and used to search titles and abstracts. Computational fluid dynamic, shear stress, flow pattens, flow velocity, and flow rate were considered as hemodynamic parameters in search terms. Animal studies were excluded. The search strategy was prepared in collaboration with a clinical librarian. The detailed search terms are provided in Appendix A. 

Titles and abstracts of all retrieved studies were independently screened by two reviewers (YS and RM) to identify potentially relevant studies. The full text of potentially eligible studies was independently assessed for eligibility by the same reviewers. Any disagreement between reviewers was resolved by consensus. In addition, all bibliographies of relevant articles and relevant review studies were searched. Study screening and selection were performed with the assistance of the webtool CADIMA (Julius Kühn-Institut, Quedlinburg, Germany) [24].

### 2.3. Data Extraction and Quality Assessment

Patient’s age, sex, number of studied IAs, method of acquiring CoW morphological information, setting of boundary conditions, hemodynamic parameters, aneurysmal state (ruptured or unruptured), and study conclusions were extracted from each study. The Risk Of Bias In Non-randomized Studies—of Exposures (ROBINS-E) tool was applied for quality assessment [25].

## 3. Results

### 3.1. Selection Flow Chart

The search yielded 1046 records. After evaluation of the titles and abstracts, 70 articles were selected for full-text assessment. Three studies met the inclusion criteria for this review [26,27,28]. The most frequent reason for exclusion (*n* = 30) was that adapted boundary conditions were not personalized. The selection progress is described in detail in Figure 2. Four studies were excluded due to their non-English full texts; they are listed in Appendix A. All three studies were case reports. Given the design of the studies, a quality assessment (ROBINS-E) was not performed.

### 3.2. Study Characteristics

The three studies were all case report in which the hemodynamic parameters of 13 aneurysms in six patients were analyzed. Eleven of these aneurysms were unruptured; the state of the other two IAs was not clarified. One study analyzed eight unruptured IAs in a 72-year-old female; in the other two studies, patients with a single IA were reported. One study did not report age and sex. Information regarding the CoW-configuration was obtained from DSA, MRA, and CTA images, respectively. Hemodynamic parameters included WSS in all cases, OSI in four patients, von-Mises tension in two patients, flow velocity in two patients, and flow rate of parent arteries in one patient. Study characteristics are listed in Table 1.

### 3.3. Personalization of Circle of Willis Hemodynamics

Jou et al. meshed patient-specific CoW-configuration and IAs morphology from DSA, and used this for three-dimensional (3D) numerical modeling [26]. Their model was composed of three flow inlets and 13 outflow branches. The outflow conditions were settled by: (1) Total mean flow rate of the CoW 12 mL/s; (2) The ratio of left/right anterior circulation and posterior circulation 1.5:1.5:1; (3) The inflow at ICAs and BA were specified to ensure each territory maintained the predetermined flow rate. ICA waveform from another patient acquired by phase-contrast MRA was applied. 

Liang et al. meshed CoW and IA from MRA images, using a one-dimensional (1D) model to simulate global hemodynamics, and thus outputting the boundary conditions of a 3D model to simulate the intra-aneurysmal flow pattern of AComA aneurysms [27]. The inlet and outlet boundary of the 3D model was the bilateral A1-segment (pre-AComA) and A2-segment (post-AComA). The inflow and outflow conditions of their 3D model were determined by a zero–dimensional model of the entire cardiovascular system. Both patient-specific and population-generic geometrical properties of the cerebral arteries were applied. Particle image velocimetry (PIV) experiments were performed with a silicon replica model (made with a 3D printing technique) to validate the 3D model on intra-aneurysmal blood flow.

Jahed et al. segmented CoW and IA from CTA images [28]. Patient-specific pulsatile velocity profiles of inlets and outlets were obtained from Transcranial Doppler (TCD). A one-way fluid–structure interaction (FSI) model was used. Elastic modulus was specifically settled in this study at 1 MPa for a 10-year-old boy and 5 MPa for a 70-year-old male. 

### 3.4. Study Findings

By comparing eight unruptured IAs in one patient, Jou et al. concluded that IA size is unrelated to blood pressure and flow rate in the parent artery [26]. Neither OSI nor any WSS-based hemodynamic variable is correlated with aneurysm size. However, when group three IAs sized between 4 mm and 6.6 mm are the large IA group, and five IAs sized between 1.5 mm and 4 mm are the small IA group, then the parent artery flow rate was linearly correlated with IA size within each size group.

Comparing three patient-specific models with models using population generic cerebral artery diameter and length, Liang reported a higher IA and parent artery average total area WSS in all three patients compared to the general population setting [27]. Two patients had higher area average OSI, while one patient had a lower OSI. Liang concluded that IA local hemodynamics were sensitive to the patient-specific treatment of boundary conditions. 

Jahed reported that, in the anterior circulation, flow velocity on the contralateral side was higher than on the IA side [28]. The maximum value of WSS and von-Misses stress was observed in the artery bifurcation and the IA neck. Their contours were consistent with the flow velocity profile.

## 4. Discussion

This systematic review identified only three case studies on simulated CoW hemodynamics in patients with IAs. Because of non-standardized hemodynamic reporting forms and different settings of the patient-specific model, it was not possible to perform a meta-analysis. It is still an open question whether simulated CoW hemodynamics are relevant to the formation, growth, and rupture of IAs. 

### 4.1. Indirect Indices

Local hemodynamics appears to be promising for predicting aneurysmal formation and rupture. For example, studies on a cellular level revealed that endothelial and smooth muscle cells respond to local hemodynamic changes through sophisticated mechanobiological mechanisms [29,30,31,32,33,34]. Chalouhi et al. showed furthermore that there is an association between changes in the cerebral blood vessel wall and IA caused by hemodynamic stress stimuli [35]. Intensive studies on local hemodynamics and IA formation or rupture have been carried out by using the CFD model. The hemodynamic indices include WSS, OSI, and von-Mises tension.

Wall shear stress was the most common parameter reported in these three studies. WSS is defined as the frictional force due to the viscosity of blood flow in the endothelium, which was most based on laminar flow assumption and determined by local velocity gradient. Due to the large velocity gradient around IAs, WSS calculation may be sensitive to near wall mesh resolution. Few of these studies mentioned the mesh convergence study in terms of WSS. The local velocity profile is mainly dominated by the flow distribution in the parent artery and the local morphology pattern. In these studies, the flow distribution among cerebral arteries was assumed to be linked with the geometrical properties of the cerebral arteries, i.e., the outflow was either predefined with cross-section related function or imposed by constant (capillary) pressure. However, there is no solid evidence indicating those assumptions can reflect personalized cerebral blood flow redistribution. It is not surprising that the WSS derived from patient-specific cerebral hemodynamics was established to be irrelevant to IA size [26]. WSS was found to be elevated at arterial bifurcations and the IA neck [28], which could potentially be attributed to a narrow neck increasing the local velocity gradient. WSS was also found to be higher around unruptured IA than in a model with general population input, implying that a high WSS might be correlated to the formation of IAs [27]. However, without a control, it cannot be concluded that high WSS is the cause of IAs. 

Oscillatory shear stress was analyzed in two studies (four patients). OSI is defined as the ratio between the magnitude of the time-averaged WSS vector and time-averaged WSS magnitude. In a patient with multiple IAs, no correlation was found between IA size and OSI [26]. In the other study, comparing patient-specific CoW hemodynamics and population generic CoW hemodynamics as the boundary condition for anterior communicating artery (AComA) IA simulation, OSI was found to be 50% higher in personalized conditions in two cases versus 34.5% lower in one case [27]. OSI essentially reflects the predominant direction of the blood flow during the cardiac cycle, which is strongly influenced by the cerebral flow distribution that is sensitive to the applied boundary condition. Hence, conclusions regarding the relationship between IAs and local OSI cannot be drawn from the above studies.

Von-Mises tension was another reported parameter. Different from WSS and OSI, which are derived from the fluid friction force on the surface of the inner vessel wall, von-Mises tension is a material parameter in solid mechanics, describing the yielding of vessel wall under complex loads by pulse pressure on vessel. Due to the coupling FSI simulation, von-Mises tension demonstrates a similar contour pattern as WSS, i.e., maximum at arterial bifurcations and at the IA neck within the CoW [28]. In addition, the thicknesses of the vessel and IAs were usually not available in the study but could significantly influence the von-Mises tension contours. Therefore, it was difficult to draw any conclusion regarding IA pathology and stress.

While researchers expressed great interest in the above-mentioned hemodynamic indices, they were not measurable in vivo with current techniques, but could only be indirectly derived from simulation. Modeling uncertainty has been a controversy; the boundary conditions, the numerical solver, and the properties of blood and vessel wall were diverse, while all have an influence on the simulation output [16,36]. One included study elaborated that AComA aneurysm regional dynamic forces were sensitive to patient-specific bilateral A1 and A2 flow conditions [27]. This finding was consistent with another study that reported that AComA aneurysm WSS distribution was sensitive to the ratio of left and right A1 flow rate [37]. A recent study compared CFD simulated dynamic forces of ICA aneurysms based on patient-specific versus generalized inflow data. It demonstrated that average WSS was sensitive to patient-specific flow rate but not to flow waveform, while OSI is less sensitive to neither flow rate nor waveform [38]. A systematic review summarized the correlation between local dynamic force and IA formation or rupture, with a meta-analysis reporting a significant mean-WSS difference between ruptured and unruptured IAs. No difference in WSS gradient and OSI was found between groups, although the meta-analysis had substantial heterogeneities [39]. It is difficult to reach conclusions on these hemodynamic indices since there is no universally accepted numerical model for cerebral simulation.

Researchers have noticed that boundary condition was one of the major causes of uncertainty in the hemodynamic model. A recent study based on 156 IAs from various locations reported that WSS and OSI are statistically different between generalized and patient-specific inflow boundary condition settings [40]. They further statistically compared those forces between rupture and unruptured IA groups, concluding that both model settings can detect the force difference between rupture and unruptured IA. This conclusion might help researchers in simplifying CFD settings. However, instead of paired *t*-test, the Mann–Whitney U test was adapted in their statistical analysis, which was not able to detect the difference between paired observations. Thus, their conclusion may be questionable. Since patient-specific boundary conditions are essential to CFD simulation accuracy, the question of whether such sensitivity, a.k.a. difference of ‘smallest detectable change’ of the two model settings, is necessary to detect the ‘minimal important change’, remains unaddressed.

### 4.2. Direct Indices

While the indirect indices were usually used to assess the local hemodynamics at the site of IAs, the direct index, i.e., flow rate, was commonly employed to evaluate the hemodynamic effect at the whole cerebral arterial network level. The configuration of the CoW has been approved having correlation with IAs. For example, epidemiological studies found that variation of A1 segments was more frequently present in the ruptured AComA aneurysm group than in the unruptured AComA aneurysm group [41]. The odds of having a ruptured AComA aneurysm in people with hypoplasia or absent A1 segment was 3.72 times of people with normal A1 segments [42]. Such correlation can be further explained from hemodynamic point of view. A hemodynamic study with phase-contrast MR found that the flow rate of the A1-segment with complete CoW was significantly lower than in the A1-segment with incomplete CoW [21]. These studies have provided statistical evidence of relationship between CoW hemodynamics and IA formation or rupture.

Flow rates were reported in two studies in this review. In the case report with multiple IAs, despite a good linear correlation between parent artery flow rate and IA size within each size group, no explanation was provided to support a grouping standard [26]. Considering its pseudo patient-specific boundary condition settings, it was hard to distinguish the finding from coincidence. In the study focused on AComA aneurysm patients, the bilateral A1 inflow rate by simulation with the patient-specific model was considerably different from that by simulation with the population-generic model [27]. This finding implied the correlation between regional flow rate and IA formation. It was consistent with a study on ruptured ICA aneurysms, in which flow rate detected by implantable electromagnetic flow probe was found lower in the ipsilateral than in the contralateral ICA [43]. In clinical practice, the flow rate in cerebral arteries is commonly altered with cerebral blood flow (CBF), defined as the blood volume that flows per unit volume per unit time in brain tissue. In a cohort of 310 Moyamoya patients, regional CBF measured with perfusion-CT was lower in patients with IAs [44]. Knowing that the flow rate of parent arteries is majorly determined by the CoW configuration, the above-mentioned findings suggested a promising association between parent artery hemodynamics and IA. Numerical tools have shown great potential in this topic. However, improvement is needed to integrate medical statistics and computer science in this interdisciplinary research field.

Unlike indirect hemodynamic indices, flow rate/velocity can (partially) be detected by modern imaging techniques, which have been widely used for numerical model validation. Two studies in this review have applied CoW flow velocity for model validation. Jahed et al. compared simulated and TCD measured left MCA flow velocity. They showed similar pulsation patterns, but simulated velocity was larger in one patient versus smaller in the other patient [28]. Liang et al. compared intra-IA velocity between PIV experiment and CFD simulation over a pulsatile cycle. A good agreement was observed at the IA edge area, while discrepancy was observed in the central region [27]. Both studies only presented a single case comparison which was insufficient to test the models’ reliability. Therefore, a quantitative validation with a sufficient sample size is needed. Sometimes, specific pathophysiological conditions, such as vasospasm, are required further for certain research circumstances [45,46].

It is necessary to clarify the importance of the role of patient specific configuration of the CoW on the cerebral hemodynamics at the end of this discussion. A typical CoW is a ring-shape arterial network with multiple afferent and efferent segments. The hemodynamics in the segments of CoW is mathematically determined by three parameters: resistance of efferent segments, inflow of afferent segments and resistance of segments in CoW. The first and second parameters are the basic model inputs, i.e., inlet and outlet boundary conditions, which are commonly selected based on additional hypothesis or vivo measurement, with large variation in the studies though. The third one is commonly neglected in most of the numerical studies of local hemodynamics of IAs. The delicate relation between the flow and the configuration of the CoW has been previously described: the flow in the segment of interest is not only influenced by its resistance, but also strongly influenced by resistance of other segments [22]. Neglecting the CoW network in the numerical models potentially introduced bias in the later analysis. Therefore, the discussion sticks on the possible influence of the CoW configuration on the commonly used indirect and direct hemodynamic indices.

## 5. Conclusions

This systematic review aimed to figure out whether hemodynamic parameters obtained from cerebral hemodynamic models correspond to IA formation, growth, or rupture. However, based on only three case reports, no firm conclusions could be drawn. A numerical model can provide comprehensive information about the cerebral blood flow as well as local flow details in IAs, which can potentially derive useful hemodynamic indices for clinical decision-making. Due to the inherent nature of a numerical model, i.e., its sensitivity to applied boundary conditions and difficulties to determine the properties of the arterial network, it is currently not recommended to apply the numerical model in patient-specific clinical studies. More sensitive analysis and rigorous validations are required to determine its measurement error and extend its application into clinical practice, especially when aiming for a personalized management in aneurysmal patient. The CoW configuration is critical on the hemodynamics in the circle. The systematic review calls for awareness from the aneurysmal hemodynamic community to reconsider the influence of CoW configuration on aneurysmal regional hemodynamics. 

## Figures and Tables

**Figure 1 jpm-12-01008-f001:**
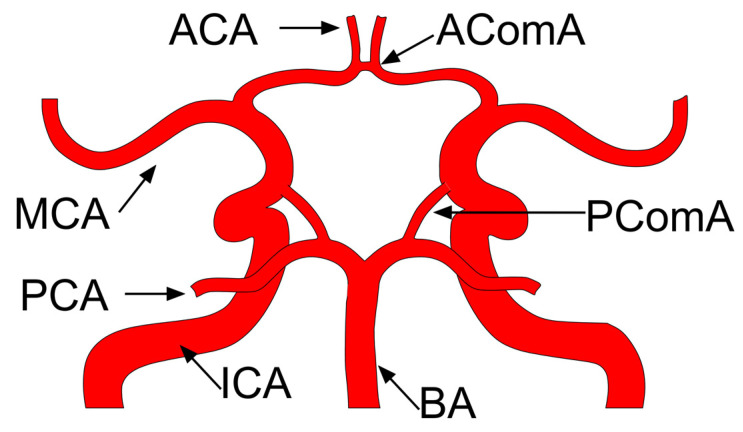
Schematic diagram of a classic circle of Willis. ACA anterior cerebral artery; MCA middle cerebral artery; PCA posterior cerebral artery; ICA internal carotid artery; BA basilar artery; AComA anterior communicating artery; PComA posterior communicating artery.

**Figure 2 jpm-12-01008-f002:**
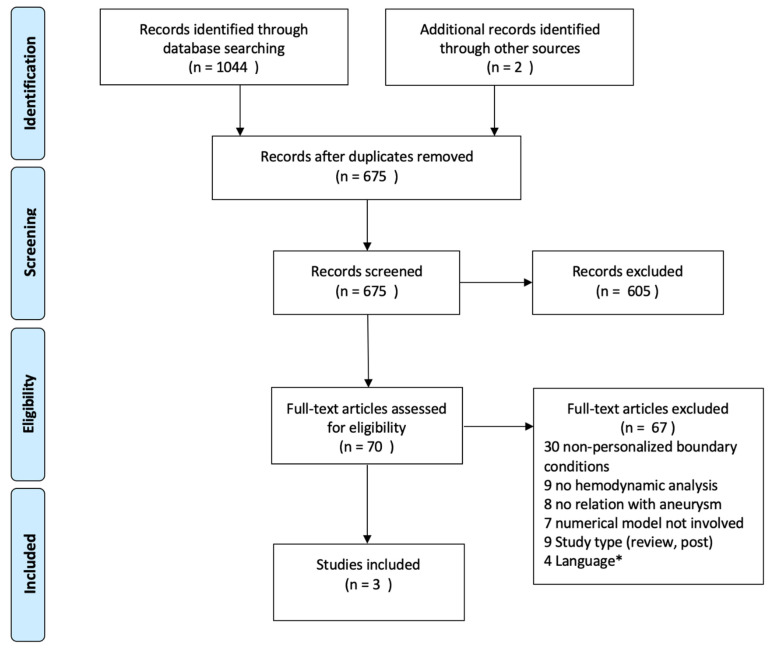
PRISMA Flow Diagram. * *List in Appendix A*.

**Table 1 jpm-12-01008-t001:** Study characteristics of three selected case-report after full-text screening.

Reference	Age	IA Number	IA State	IA Location	Measured Method	Hemodynamic Parameter	Main Findings and Conclusions
[26]	72 y	8	unruptured	ICA, MCA, BA tip, AComA, PComA	DSA + CFD	Flow rate, WSS, OSI	Wall shear stress-derived hemodynamic parameters are not related to aneurysm size.
[27]	Unknown	3	unruptured	AComA	MRA + 1D-3D Model	Flow rate, Flow velocity, WSS, OSI	Boundary conditions of the aneurysm models played an important role in simulated intra-aneurysmal flow patterns.
[28]	10 y/70 y	2	unknown	MCA, BA tip	TCD + CTA + CFD	Flow velocity, WSS, Von mises tension	The distribution of stress was dependent on the geometry of the circle of Willis.

IA intracranial aneurysm; ICA internal carotid artery; MCA middle cerebral artery; BA basilar artery; AComA anterior communicating artery; PComA posterior communicating artery; DSA Digital Subtraction Angiography; MRA magnetic resonance angiography; CTA computed tomography angiography; CFD Computational fluid dynamic; 1D-3D Model hydride numerical model of one dimension and three dimensions; WSS wall shear stress; OSI Oscillatory shear index.

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
