# Peer review of "The Role of Hemodynamics through the Circle of Willis in the Development of Intracranial Aneurysm: A Systematic Review of Numerical Models"

_jpm, 2022, doi:10.3390/jpm12061008_

Round 1

Reviewer 1 Report

The authors have conducted a systematic review on whether simulated hemodynamic indices of the circle of Willis (CoW) are relevant to the formation, growth, or rupture of intracranial aneurysm. They followed PRISMA guidelines in this review.

Although the paper is well written, its value as a systematic has diminished due to just 3 papers. Here are a few minor changes that I have:

-Table 1: Please add one more column at the end and provide a summary of findings and conclusions. Column heading can be "Main Findings & Conclusions".

-Tables are self-containing and stand alone. A reader should understand the information presented in the table without referring to the text of the paper. Please give more footnotes at the bottom of the table (type of studies, expansion of acronyms, statistics used, etc.) 

-Table 1: The first column should have a column heading "Reference".  Under that column, give only the reference number like [25], [26], etc. There is no need to give authors names.

-Add a brief search strategy in the main manuscript itself instead of placing it in the Supplement file.

Author Response

Dear reviewer,

Thank you for your insightful comments. Please find our responses in the attachment.

Sincerely,

Yuanyuan Shen

Reviewer 2 Report

Shen et al. present a systematical review on the numerical models that might help to explain the circle of Willis intracranial aneurysms. Although the aim of the review is valid, my greatest concern is regarding the number of papers that met the inclusion criteria (only three case reports). On that thought, I have some comments/questions directed to the authors:

1. The three articles that met the inclusion criteria are quite recent (2016 and 2018). But it is not mentioned if the articles were filtered regarding the year of publication. Nevertheless, in the Introduction it is mentioned the hemodynamics model has existed for decades.

2. I understood the authors used the search strategy "NOT "Humans"[Mesh]" to avoid experimental studies; however, I am concerned this criteria might have led to the exclusion of some studies that could be important in the review. Maybe using the term "in vitro" as an exclusion criteria would have performed better for the authors' intentions.

3. Suggestion: image of the Circle of Willis with the name of the arteries would be interesting to the manuscript.

4. Minor comment: In line 55-56, when the authors mention that meta-analysis based on the computational models were found unreliable, there is no reference cited.

Author Response

Dear reviewer,

Thank you for your insightful comments. Please find our responses in the attachment.

Sincerely

Yuanyuan Shen

Round 2

Reviewer 2 Report

The authors have properly addressed my concerns. I believe the manuscript is fit for acceptance in the Journal of Personalized Medicine.